# NT-proBNP testing for heart failure diagnosis in people with atrial fibrillation: A diagnostic accuracy study

Nicholas R. Jones[1]*, Kathryn S. Taylor[1], José M. Ordóñez-Mena[1], Clare R. Goyder[1], F. D. Richard Hobbs[1], Clare J. Taylor[1,2]*

1 Nuffield Department of Primary Care Health Sciences, University of Oxford, Oxford, United Kingdom,
2 Department of Applied Health Sciences, University of Birmingham, Birmingham, United Kingdom

* nicholas.jones2@phc.ox.ac.uk (NRJ); c.j.taylor.1@bham.ac.uk (CJT)

## Abstract

### Background

N-terminal pro-B-type natriuretic peptides (NT-proBNP) are important in the assessment of suspected heart failure (HF). However, NT-proBNP concentrations are elevated in atrial fibrillation (AF), creating diagnostic uncertainty. The aim of this study was to assess the diagnostic accuracy of NT-proBNP for HF in people with AF, overall and by age, sex and BMI.

### Methods and findings

Retrospective study of all patients with a NT-proBNP test in their primary care electronic health record among English GP practices provided through the Clinical Practice Research Datalink (2004–2018) and linked to secondary care data. The accuracy of NT-proBNP for diagnosing HF within six months was assessed for people with and without AF at thresholds of 125, 400, 660 and 2,000 pg/mL, including by age, sex and BMI. Among 155,347 people who had an NT-proBNP test organized in primary care (median age 61 years), 17,403 (11.2%) had pre-existing AF. Of the 155,347 people included, 14,585 (9.4%) were subsequently diagnosed with HF, including 4,168/17,403 (23.9%) people with AF (median NT-proBNP = 1,852 pg/mL, interquartile range (IQR) [974, 3,459] pg/mL) and 10,417/137,944 (7.6%) without AF (1,110 pg/mL, IQR [434, 3,108] pg/mL). NT-proBNP discriminated better overall among people without AF (AUC = 0.877 (95% confidence interval (CI) [0.873, 0.881]) than with AF (AUC = 0.743 (95% CI [0.735, 0.751]). Among people with AF, NT-proBNP sensitivity and specificity at a 125 pg/mL threshold was 98.8% (95% CI [98.5%, 99.1]) and 13.2% (95% CI [12.6%, 13.7]) and at 400 pg/mL 93.2% (95% CI [92.4, 93.9]) and 35.5% (95% CI [34.7, 36.3]). Among people without AF the corresponding results were 92.9% (95% CI [92.4, 93.4]) and 53.8% (95% CI [53.6, 54.1])

**Data availability statement:** Data for this study were obtained on license from CPRD and cannot be shared due to the contract agreements for the data access. However, equivalent data can be obtained directly from CPRD with relevant ISAC approval. Applications to use CPRD data can be made through the following website: https://www.cprd.com/. Enquiries to CPRD can be made at enquiries@cprd.com. We obtained Hospital Episode Statistics and Office for National Statistics data through CPRD linkage and these cannot be shared for the same reason. Further information on Hospital Episode Statistics, including how to request access to data is available at the following website: https://digital.nhs.uk/services/hospital-episode-statistics#accessing-data. Further information regarding the Office for National Statistics and their Secure Research Service is available at the following website: https://www.ons.gov.uk/. The code used for this analysis is freely available through GitHub at: https://github.com/nicholasjones14/DiagnoseNP_AF.

**Funding:** The original DIAGNOSE-NP study was funded by the National Institute for Health and Care Research (NIHR) Collaboration for Leadership in Applied Health Research and Care (CLAHRC) Oxford (CJT), grant number P2-115. The CLAHRC has now closed, replaced by the Applied Research Collaboration (https://www.arc-oxtv.nihr.ac.uk/). This work was also supported by the NIHR Community Healthcare Medtech and In Vitro Diagnostics Cooperative (MIC) at Oxford Health NHS Foundation Trust (https://www.community.healthcare.mic.nihr.ac.uk/) as part of the long-term condition theme (CJT and FDRH).The funders had no role in study design, data collection and analysis, decision to publish, or preparation of the manuscript.

**Competing interests:** I have read the journal's policy and the authors of this manuscript have the following competing interests. NRJ reports consultancy fees from Oxon Epidemiology outside the submitted work. NRJ and CRG are supported by University of Oxford NIHR Academic Clinical Lectureships. FDRH acknowledges support from NIHR ARC OTV and the NIHR Oxford BRC. CJT reports personal fees and non-financial support from Astra Zeneca, Roche and Bayer outside the submitted work.

at 125 pg/mL and 77.1% (95% CI [76.3, 77.9]) and 84.9% (95% CI [84.7, 85.1]) at 400 pg/mL. NT-proBNP discriminated less well among people with AF aged ≥65 years compared to <65years (e.g., AUC in people aged 65–75 years was 0.725, 95% CI [0.712, 0.739]). Increasing the threshold for a positive test among people with AF from 125 pg/mL to 660 pg/mL would reduce the number of false positive results by 26.0%, whilst retaining a negative predictive value of 91.5 (95% CI [90.8, 92.1]), albeit with a 10.6% increase in the proportion of those tested with AF having a missed or delayed HF diagnosis. The main limitation of the study is that it relies on routinely collected primary care data and people with an NT-proBNP result <400 pg/mL may not have been referred for further assessment, impacting upon the diagnostic accuracy below this threshold.

## Conclusions

NT-proBNP discriminates more accurately for HF among people without AF than with AF. A higher referral threshold could be considered in AF to account for higher median NT-proBNP levels but this would also increase missed HF diagnoses.

## Author summary

### Why was this study done?

- NT-proBNP is a blood test that GPs use to help identify people with suspected heart failure based on their symptoms and risk factors.

- People with an elevated NT-proBNP level will be referred to a hospital specialist for further testing to confirm a heart failure diagnosis, whilst people with normal NT-proBNP will be assessed by the GP for alternative causes of their symptoms.

- Atrial fibrillation and heart failure frequently co-exist and people with atrial fibrillation typically have higher levels of NT-proBNP but there is uncertainty as to how this should be reflected in threshold levels for referral.

### What did the researchers do and find?

- We used a large database of general practice records to assess how many people who had a NT-proBNP test went on to be diagnosed with heart failure.

- The analysis compared the accuracy of NT-proBNP for identifying people with heart failure, comparing between those with and without atrial fibrillation and between sexes, different age groups and categories of body mass index.

- We found that NT-proBNP levels are higher in those with atrial fibrillation and it performs less well as a test, so a higher referral threshold may be recommended (660 pg/mL) compared to those without atrial fibrillation (125 pg/mL).

The other authors have declared that no competing interests exist.

**Abbreviations:** AF, atrial fibrillation; BNP, B-type natriuretic peptide; HF, heart failure; LVEF, left ventricular ejection fraction; NPV, negative predictive value; NT-proBNP, N-terminal pro-B-type natriuretic peptide; PPV, positive predictive value.

- This would reduce the number of false positive NT-proBNP results in people with AF by 26%, whilst retaining the negative predictive value >90%, albeit with a 10% increase in the number of people with AF receiving a false negative result.

## What do these findings mean?

- Clinicians will still need to use their clinical judgement when interpreting NT-proBNP but our results could help inform the referral thresholds used in daily practice.

- The study's main limitation is it relies on what has been coded in patient's electronic health records so it is unclear if people were in atrial fibrillation at the time of NT-propBNP testing or what type of heart failure people went on to develop.

## Introduction

N-terminal pro-B-type natriuretic peptide (NT-proBNP) testing is a central part of the heart failure (HF) diagnostic pathway [1–3]. Patients with suspected HF should have an NT-proBNP test and those with an elevated result should be referred for an echocardiogram and specialist assessment to confirm the diagnosis. NT-proBNP is recommended in favor of B-type natriuretic peptide (BNP) as a more stable molecule with levels that are not affected by treatments such as sacubitril/valsartan [4]. In the outpatient and primary care setting, the European Society of Cardiology (ESC) and American Heart Association (AHA) recommend using a NT-proBNP threshold of 125 pg/mL for a positive test, whereas in the United Kingdom (UK), the National Institute for Health and Care Excellence (NICE) recommend using a threshold of 400 pg/mL [1–3].

However, NT-proBNP levels are affected by conditions other than HF. Atrial fibrillation (AF) can elevate NT-proBNP levels, even in the absence of HF. Median natriuretic peptide (NP) levels are three-times higher among people with AF compared to those in sinus rhythm [5]. Other causes of an elevated NT-proBNP include increasing age, kidney disease and valvular heart disease whilst common causes of suppressed NT-proBNP levels include increasing body mass index (BMI) and certain medications [6]. This can create uncertainty around the significance of a raised NT-proBNP level in patients with AF and consequent uncertainty regarding which patients to refer for assessment of HF. HF occurs frequently among people with AF making this a common diagnostic dilemma. A recent Danish population study reported that the lifetime risk of developing HF among people with AF from age 45 years onwards was 41.2% (95% CI [39.8, 42.7]) [7]. Similarly, a meta-analysis of nine cohort studies reported that the relative risk (RR) of HF was almost five times higher among people with AF compared to people without AF (RR 4.62, 95% CI [3.13, 6.83]) [8].

A 2023 ESC HF Association position statement on NT-proBNP testing in HF recommends the NT-proBNP threshold should be increased by 50% where the ventricular rate is ≤90 beats per minute or by 100% if the ventricular rate is above this [4]. The ESC also recommends different thresholds to support a diagnosis of HF with preserved ejection fraction (HFpEF) for people with AF compared to sinus

rhythm [5]. However, most international guidelines do not yet recommend using adjusted NT-proBNP thresholds for people with AF when considering which patients should be referred for specialist assessment of suspected HF from primary care, although clinicians are recommended to take patient factors into account when interpreting the NT-proBNP result [1–3]. The latest NICE HF guidelines make an explicit recommendation for further research to help determine the optimal NT-proBNP threshold for the diagnosis of HF in people with AF [2]. The aim of this study was to report the performance of NT-proBNP in diagnosing HF at key thresholds, comparing patients with and without AF, overall and by age, sex and BMI.

## Methods

### Ethics statement

The DIAGNOSE-NP protocol was approved by the Independent Scientific Advisory Committee (ISAC) of the Medicines and Healthcare products Regulatory Agency (MHRA) (ISAC protocol number: 19_136; available from the authors on request). Ethics approval for observational research using the Clinical Practice Research Datalink (CPRD) with approval from ISAC was granted by a National Research Ethics Service committee (Trent MultiResearch Ethics Committee, reference number: 05/MRE04/87). Ethics approval for observational research using CPRD has previously been granted by a National Research Ethics Service committee (Trent MultiResearch Ethics Committee, reference number: 05/MRE04/87). Individual patients in the dataset do not provide explicit consent for their data to be used for research purposes but can choose to opt out of research through the national data opt out process, as outlined at https://digital.nhs.uk/services/national-data-opt-out.

### Study design

This analysis was part of a wider project (DIAGNOSE-NP) reporting on a retrospective primary care cohort of patients within the Clinical Practice Research Datalink (CPRD) GOLD and Aurum databases who had undergone NP testing [9–11]. An application was submitted to CPRD to access the data but we did not pre-publish a study protocol. The combined databases contain the patient records of >15% of the UK population and have been shown to be representative of the wider population [12]. Patient records were linked to inpatient Hospital Episodes Statistics (HES) and the Index of Multiple Deprivation (IMD) socioeconomic data, which limited the study to England.

### Participants

Patients were included in this analysis if they were aged ≥45 years and had a NT-proBNP test result recorded in their primary care electronic health record between 1 January 2004 and 31 December 2018. Patients also needed to be registered at an 'up-to-standard' GP practice (a CPRD quality measure) for a minimum of 12 months and be eligible for data linkage. Patients were excluded if they had been diagnosed with HF before the study index date or before the date of the first recorded NT-proBNP test.

This analysis was restricted to the diagnostic accuracy of NT-proBNP because this is the test recommended by NICE and the ESC [1,2]. NT-proBNP is recommended to assess for heart failure among people with symptoms such as breathlessness, peripheral oedema or fatigue. NT-proBNP is used as a 'rule out' test in primary care, because patients without an elevated result are generally not referred on for further assessment of heart failure. Symptomatic patients with an elevated NT-proBNP should be referred for a specialist assessment to confirm or refute the diagnosis of heart failure, including further tests such as an echocardiogram. NT-proBNP thresholds therefore need to be set at a level that prioritizes high sensitivity and negative predictive value (NPV) to minimize false negative results so that true cases of heart failure are not missed. At an individual patient level, low specificity is less problematic as individuals will proceed to specialist tests such as echocardiogram to confirm a diagnosis of heart failure but this have capacity implications for finite echocardiography services.

## Test methods

Patients entered the cohort on the date of their NT-proBNP test (index test). The primary outcome was a diagnosis of HF (reference standard) within six months of the most recent NT-proBNP test, recorded in either their primary care (CPRD) or secondary care (HES) record. Absence of a HF code was assumed to mean absence of a diagnosis. Patients exited the cohort on the date they were diagnosed with HF or six months after the NT-proBNP test date for patients who were not diagnosed with HF.

The clinical codes used to identify NT-proBNP tests, AF and HF diagnosis were derived from the Quality and Outcomes Framework guidance and the National Health Service (NHS) terminology and classifications browser (S1 Appendix). Diagnoses of HF made in primary care were validated through data linkage with HES using International Classification of Diseases, 10th revision codes.

## Analysis

The sociodemographic details of the cohort were summarized using median and interquartile range (IQR) for continuous variables, and frequencies and percentages for categorical variables. These were analyzed comparing between patients with and without AF and whether or not patients were diagnosed with HF.

NT-proBNP was included as both a continuous variable and a categorical variable at the AHA and ESC (≥125 pg/mL) and NICE (≥400 pg/mL) referral thresholds and the NICE urgent referral threshold of ≥2,000 pg/mL. In contrast to our previous analyses of this dataset, within this study we include an analysis at a threshold of 660 pg/mL because this reflects the value the ESC recommend as a major criterion for diagnosing HFpEF among people with AF [1–3,5]. NICE recommend people with an NT-proBNP ≥ 2,000 pg/mL are seen within two weeks of referral and people with an NT-proBNP ≥ 400 pg/mL are seen within six weeks so we report on the proportion of HF diagnoses made within those timeframes. The diagnostic accuracy of NT-proBNP for HF diagnosis was tested by calculating sensitivity, specificity, positive predictive value (PPV) and NPV, likelihood ratio, and diagnostic odds ratio using the 'epitools' package [13]. The 95% confidence intervals (CIs) were calculated using the binomial distribution for proportions and using the Wald's normal approximation for ratios. Receiver operating characteristic (ROC) curves were plotted separately for patients with or without AF to allow for comparison of overall test performance between patient groups. The area under the ROC curve (AUC) was estimated using the 'pROC' package [14].

Sub-group analyses were undertaken comparing by sex, age and category of BMI. Age was categorized as 45–64, 65–74 and ≥75 years, reflecting the age bands used in existing AF risk-prediction scores such as $CHA_2DS_2VASc$ [15]. BMI was categorized according to World Health Organization categories; except we used a BMI of <20 kg/m$^2$ to define underweight as there were limited data for people with AF and a BMI < 18.5 kg/m$^2$ to undertake an analysis at that threshold [16]. ROC curves were plotted by categories of age (<65, 65–75, >75 years) and BMI comparing healthy weight to overweight and obese among people with and without AF.

All analyses were carried out using R (version 4.4.0). This study is reported as per the Standards for Reporting of Diagnostic Accuracy (STARD) guideline (S1 Checklist).

## Results

### Heart failure diagnosis

A total of 155,347 patients had an NT-proBNP test during the study period, including 17,403 (11.2%) patients with an existing diagnosis of AF and 137,944 (88.8%) patients without existing AF (Fig 1). Among those tested, 14,585 (9.4%) people were diagnosed with HF within the subsequent six months, including 4,168 (23.9%) of those with AF and 10,417 (7.6%) of people without AF. Among HF cases, the median time between NT-proBNP test and diagnosis was 26 days (IQR 7–64 days). Time between NP testing and a HF diagnosis was similar between those with and without AF.

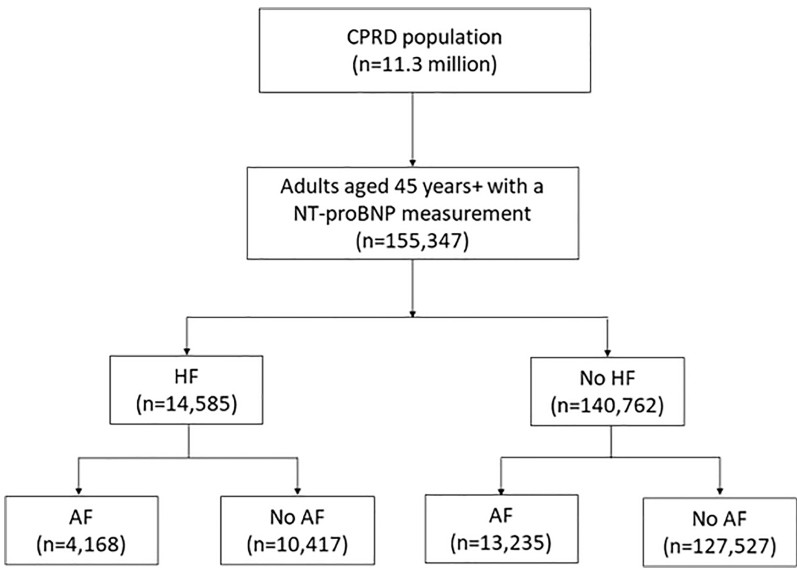

**Fig 1. Flow chart of participants included in the study, categorized by presence of heart failure and atrial fibrillation.**

Overall, the median NT-proBNP level among all patients diagnosed with HF was 1,358 pg/mL (IQR [534, 3,230] pg/mL). Among those with AF the median NT-proBNP was 1,852 pg/mL (IQR [974, 3,459] pg/mL) compared to 1,110 pg/mL (IQR [434, 3,108] pg/mL) among people without AF.

The median NT-proBNP among people who were not subsequently diagnosed with HF was 124 pg/mL (IQR [56, 306] pg/mL), with median levels of 724 pg/mL (IQR [245, 1,535] pg/mL) among people with AF and 111 pg/mL (IQR [51, 250] pg/mL) among people without AF.

## Population characteristics by presence of AF

Among those tested, the overall median age was 61 years but people with AF tended to be older (median age 66.2 years (IQR [60.0, 72.9] years) versus 60.0 years (IQR [51.0, 68.3] years)) and a higher proportion were male (51.5% versus 41.3%), of white ethnicity (96.5% versus 90.5%) and in the least deprived socio-economic quintile (22.4%) versus 19.0%) (Table 1). Those with AF were also more likely to have hypertension, ischemic heart disease, a prior stroke, valvular heart disease or other cardiovascular disease (Table 1). Among those who were subsequently confirmed to have a diagnosis of HF, a higher proportion of people also had these co-morbidities among the population with AF compared to those without AF (Table 1).

The left ventricular ejection fraction (LVEF) was only recorded in 836 individuals, 595 of whom were diagnosed with HF, which was too small a proportion to undertake an analysis on this basis. Where reported, the median LVEF was 49% (IQR [35, 58]%) among people with AF diagnosed with HF compared to 43% (IQR [30, 55]%) in those without AF who were diagnosed with HF.

## Diagnostic test accuracy parameters

NT-proBNP had higher discrimination in identifying patients with HF among people without AF (AUC = 0.877, 95% CI [0.873, 0.881]), compared to with AF (AUC = 0.743, 95% CI [0.735, 0.751]) (Fig 2). At each pre-specified NT-proBNP threshold, the sensitivity was slightly higher among patients with AF but the specificity significantly lower when compared to the corresponding results among people without AF (Fig 2). For example, at the 125 pg/mL threshold the sensitivity and specificity of

**Table 1. Baseline characteristics of patients undergoing NT-proBNP testing, by presence or absence of atrial fibrillation and heart failure diagnosis.**

| | With atrial fibrillation | | Without atrial fibrillation | |
| --- | --- | --- | --- | --- |
| | Diagnosed with HF (n = 4,168) | Not diagnosed with HF (n = 13,235) | Diagnosed with HF (n = 10,417) | Not diagnosed with HF (n = 127,527) |
| **Characteristic** | | | | |
| **Age, years, median (IQR)** | 68 (62–74) | 66 (59–72) | 66.6 (58.5–73.3) | 60 (51–68) |
| **Sex, female n (%)** | 1,918 (46.02) | 6,525 (49.3) | 5,265 (50.54) | 75,756 (59.4) |
| **Ethnicity, n (%)** | | | | |
| White | 4,049 (97.14) | 12,746 (96.31) | 9,789 (93.97) | 115,077 (90.24) |
| Asian or Asian British | 37 (0.88) | 151 (1.15) | 240 (2.31) | 4,803 (3.76) |
| Black, Black British, Caribbean or African | 23 (0.56) | 121 (0.91) | 161 (1.54) | 3,408 (2.67) |
| Mixed, multiple ethnic groups or other ethnic group | 27 (0.64) | 97 (0.74) | 96 (0.92) | 1,782 (1.4) |
| Missing | 32 (0.77) | 120 (0.91) | 131 (1.26) | 2,457 (1.93) |
| **BMI (kg/m²)** | 27.5 (24.3–31.6) | 28.1 (24.8–32.3) | 27.8 (24.3–32.2) | 28.8 (25.3–33.1) |
| **Smoking status, n(%)** | | | | |
| Never | 1,247 (29.92) | 4,197 (31.71) | 2,915 (27.98) | 42,270 (33.15) |
| Former | 2,453 (58.85) | 7,569 (57.19) | 5,647 (54.21) | 64,045 (50.22) |
| Current | 465 (11.16) | 1,449 (10.95) | 1,823 (17.5) | 20,954 (16.43) |
| Missing | 3 (0.07) | 20 (0.15) | 32 (0.31) | 258 (0.2) |
| **IMD, quintile, n(%)** | | | | |
| Q1 (least deprived) | 935 (22.43) | 2,955 (22.33) | 1,955 (18.77) | 24,240 (19.01) |
| Q2 | 997 (23.92) | 3,146 (23.77) | 2,234 (21.45) | 26,484 (20.77) |
| Q3 | 867 (20.8) | 2,854 (21.56) | 2,150 (20.64) | 25,722 (20.17) |
| Q4 | 732 (17.56) | 2,386 (18.03) | 2,106 (20.22) | 25,963 (20.36) |
| Q5 (most deprived) | 633 (15.19) | 1,881 (14.21) | 1,961 (18.82) | 25,052 (19.64) |
| Missing IMD | 4 (0.1) | 13 (0.1) | 11 (0.11) | 66 (0.05) |
| **SBP (mmHg)** | 133 (122–142) | 134 (123–142) | 136 (125–146) | 136 (126–145) |
| **DBP (mmHg)** | 76 (70–81) | 77 (70–82) | 76 (69–82) | 78 (70–83) |
| **Total cholesterol (mmol/L)** | 4.3 (3.6–5) | 4.4 (3.7–5.2) | 4.54 (3.8–5.4) | 4.9 (4.1–5.7) |
| **NT-pro BNP (pg/mL)** | 1,852 (974–3,459) | 724 (245–1,535) | 1,110 (434–3,102) | 111 (51–250) |
| **Medical history, n (%)** | | | | |
| Diabetes | 1,123 (26.94) | 3,586 (27.09) | 3,064 (29.41) | 32,113 (25.18) |
| Hypertension | 3,008 (72.17) | 9,144 (69.09) | 6,857 (65.83) | 72,613 (56.94) |
| Atrial fibrillation | 4,168 (100) | 13,235 (100) | – | – |
| Angina | 612 (14.68) | 1,836 (13.87) | 1,350 (12.96) | 10,644 (8.35) |
| Ischemic heart disease | 807 (19.36) | 2,393 (18.08) | 1,777 (17.06) | 13,030 (10.22) |
| Myocardial infarction | 474 (11.37) | 1,103 (8.33) | 1,222 (11.73) | 6,930 (5.43) |
| Stroke | 750 (17.99) | 2,070 (15.64) | 1,113 (10.68) | 8,696 (6.82) |
| Valvular disease | 423 (10.15) | 1,248 (9.43) | 520 (4.99) | 3,498 (2.74) |
| Other cardiovascular disease | 1,227 (29.44) | 3,799 (28.7) | 1,849 (17.75) | 14,563 (11.42) |
| **Time between NP test and HF diagnosis (days)** | 23 (7–60) | 676 (371–1,219) | 27 (7–66) | 837 (438–1,429) |
| <2 weeks | 1,611 (38.65) | – | 3,748 (35.98) | – |
| 2-6 weeks | 1,093 (26.22) | – | 2,606 (25.02) | – |
| ≥6 weeks | 1,464 (35.12) | – | 4,063 (39) | – |

Abbreviations: BMI, body mass index; BNP, B-type natriuretic peptide; DBP, diastolic blood pressure; HF, heart failure; IMD, Index of Multiple Deprivation; IQR, interquartile range (25th and 75th percentiles); NP, natriuretic peptide; Q, quintile; SBP, systolic blood pressure.

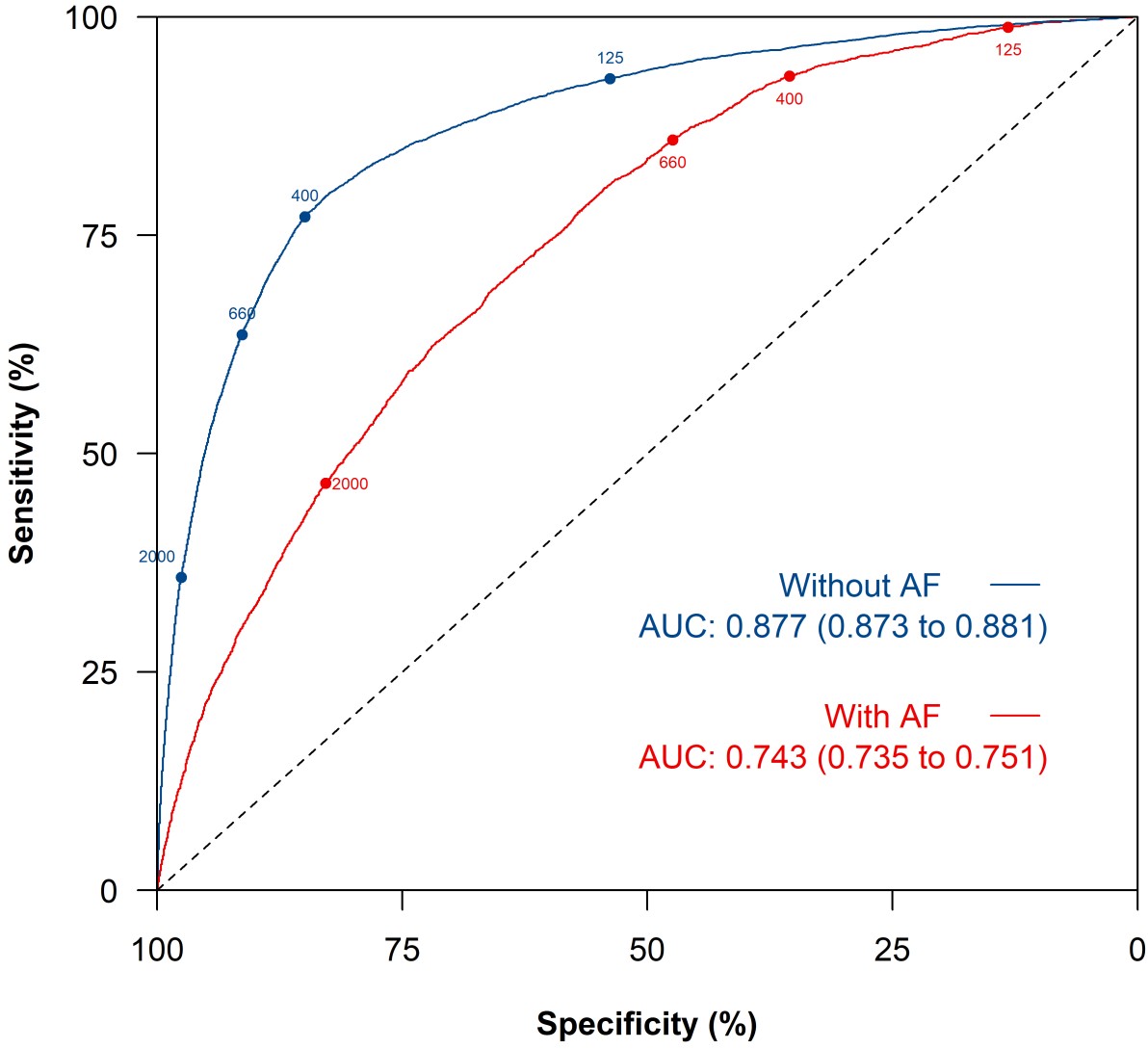

**Fig 2. Receiver operating characteristic curve of NT-proBNP for heart failure diagnosis among people with and without atrial fibrillation (AF).**
Legend: Figure shows the key referral and diagnostic thresholds in current international guidelines. AUC, area under the curve with 95% confidence interval reported in brackets.

NT-proBNP among people with AF were 98.8% (95% CI [98.5, 99.1]) and 13.2% (95% CI [12.6, 13.7]), respectively compared to corresponding values among people without AF of 92.9% (95% CI [92.4, 93.4]) and 53.8% (95% CI [53.6, 54.1]) (Table 2). At the same threshold, the PPV and NPV among people with AF was 26.4% (95% CI [25.7, 27.1]) and 97.3% (95% CI [96.5, 98.0]) compared to 14.1% (95% CI [13.9, 14.4]) and 98.9% (95% CI [98.9, 99.0]) among people without AF.

At a 400 pg/mL threshold in people with AF the sensitivity was 93.2% (95% CI [92.4, 93.9]), specificity 35.5% (95% CI [34.7, 36.3]) and NPV 94.3% (95% CI [93.6, 94.9]). At the same threshold among people without AF the sensitivity, specificity and NPV were 77.1% (95% CI [76.3, 77.9]), 84.9% (95% CI [84.7, 85.1]) and 97.8% (95% CI [97.8, 97.9]), respectively. Even at a threshold of 660 pg/mL among people with AF the positive likelihood ratio (LR + 1.63, 95% CI [1.60, 1.67]) was lower than that achieved at a threshold of 125 pg/mL in people without AF (LR + 2.01, 95% CI [2.00, 2.03]), whilst the negative likelihood ratios remained similar.

**Table 2. Diagnostic test accuracy parameters for the diagnosis of HF using NT-proBNP level at NICE and ESC referral thresholds for those with and without atrial fibrillation.**

| | With atrial fibrillation (n = 17,403) | | | | Without atrial fibrillation (n = 137,944) | | | |
|---|---|---|---|---|---|---|---|---|
| Overall prevalence of heart failure, % (95% CI) | 23.9 (23.3–24.6) | | | | 7.6 (7.4–7.7) | | | |
| NT-proBNP threshold (pg/mL) | ≥125 | ≥400 | ≥660 | ≥2,000 | ≥125 | ≥400 | ≥660 | ≥2,000 |
| TP, n | 4,120 | 3,884 | 3,582 | 1,944 | 9,681 | 8,029 | 6,629 | 3,730 |
| FN, n | 48 | 284 | 586 | 2,224 | 736 | 2,388 | 3,788 | 6,687 |
| FP, n | 11,494 | 8,540 | 6,959 | 2,278 | 58,871 | 19,248 | 11,039 | 3,146 |
| TN, n | 1,741 | 4,695 | 6,276 | 10,957 | 68,656 | 108,279 | 116,488 | 124,381 |
| Sensitivity, % (95% CI) | 98.8 (98.5, 99.1) | 93.2 (92.4, 93.9) | 85.9 (84.8, 87.0) | 46.6 (45.1, 48.2) | 92.9 (92.4, 93.4) | 77.1 (76.3, 77.9) | 63.6 (62.7, 64.6) | 35.8 (34.9, 36.7) |
| Specificity, % (95% CI) | 13.2 (12.6, 13.7) | 35.5 (34.7, 36.3) | 47.4 (46.6, 48.3) | 82.8 (82.1, 83.4) | 53.8 (53.6, 54.1) | 84.9 (84.7, 85.1) | 91.3 (91.2, 91.5) | 97.5 (97.4, 97.6) |
| PPV % (95% CI) | 26.4 (25.7, 27.1) | 31.3 (30.4, 32.1) | 34.0 (33.1, 34.9) | 46 (44.5, 47.6) | 14.1 (13.9, 14.4) | 29.4 (28.9, 30.0) | 37.5 (36.8, 38.2) | 54.2 (53.1, 55.4) |
| NPV % (95% CI) | 97.3 (96.5, 98.0) | 94.3 (93.6, 94.9) | 91.5 (90.8, 92.1) | 83.1 (82.5, 83.8) | 98.9 (98.9, 99.0) | 97.8 (97.8, 97.9) | 96.9 (96.8, 96.9) | 94.9 (94.8, 95.0) |
| LR+ (95% CI) | 1.14 (1.13, 1.15) | 1.44 (1.42, 1.47) | 1.63 (1.60, 1.67) | 2.71 (2.58, 2.85) | 2.01 (2.00, 2.03) | 5.11 (5.02, 5.19) | 7.35 (7.18, 7.52) | 14.5 (13.9, 15.2) |
| LR− (95% CI) | 0.09 (0.07, 0.12) | 0.19 (0.17, 0.22) | 0.30 (0.27, 0.32) | 0.64 (0.63, 0.66) | 0.13 (0.12, 0.14) | 0.27 (0.26, 0.28) | 0.40 (0.39, 0.41) | 0.66 (0.65, 0.67) |
| DOR (95% CI) | 12.96 (9.82, 17.54) | 7.51 (6.64, 8.54) | 5.51 (5.02, 6.06) | 4.20 (3.90, 4.54) | 15.34 (14.23, 16.55) | 18.91 (18.03, 19.85) | 18.47 (17.66, 19.32) | 22.05 (20.9, 23.28) |

**Abbreviations:** DOR, diagnostic odds ratio, FN, false negatives, FP, false positives, LR, likelihood ratio, N, number, NPV, negative predictive value, PPV, positive predictive value, TN, true negatives, TP, true positives.

### Sub-group analyses by age, BMI and sex

Discrimination of NT-proBNP remained superior among people without AF than with AF when comparing by age groups (Figs 3 and 4). The presence of AF impacted on the diagnostic accuracy of NT-proBNP to a greater extent than age (Figs 3 and 4) but overall NP testing has very poor discrimination for HF among people with AF who are 65 years or older. For example, among people with AF, the sensitivity and specificity of NT-proBNP at the ESC recommended 125 pg/mL threshold was 98.5% (95% CI [97.8, 99.1]) and 18.8% (95% CI [17.8, 19.8]), respectively among people aged 45–64 years and 99.0% (95% CI [98.4, 99.4]) and 9.7% (95% CI [8.9, 10.6]), respectively among people aged 65–74 years (Table 3). Prevalence of AF was higher among the oldest age category, meaning the PPV of a raised NT-proBNP increased with age, with a corresponding reduction in the NPV (Table 3). For example, using a threshold of 125 pg/mL the PPV and NPV among people with AF aged 45–64 years were 23.0% (95% CI [22.0, 24.1]) and 98.1% (95% CI [97.1, 98.8]), respectively, whereas for people aged ≥75 years the corresponding results were 30.4% (95% CI [28.8, 32.1]) and 93.4% (95% CI [87.5, 97.1]).

NT-proBNP performed less well as a rule-out test among younger people without AF. For example, in people aged < 65 years at a threshold of 400 pg/mL sensitivity was 70.0% (95% CI [68.7, 71.4]) and at a threshold of 125 pg/mL the sensitivity was 88.8% (95% CI [87.8, 89.7]), although the NPV remained high (98.3%, 95% CI [98.2, 98.4] at 400 pg/mL threshold) (S1 Table).

Comparing between categories of BMI, NT-proBNP had the same pattern of higher discrimination for HF among people without AF compared to with AF (S1 and S2 Figs). There was no statistically significant difference in the sensitivity or NPV of NT-proBNP at any of the diagnostic thresholds when comparing between categories of BMI among people with AF

PLOS Medicine

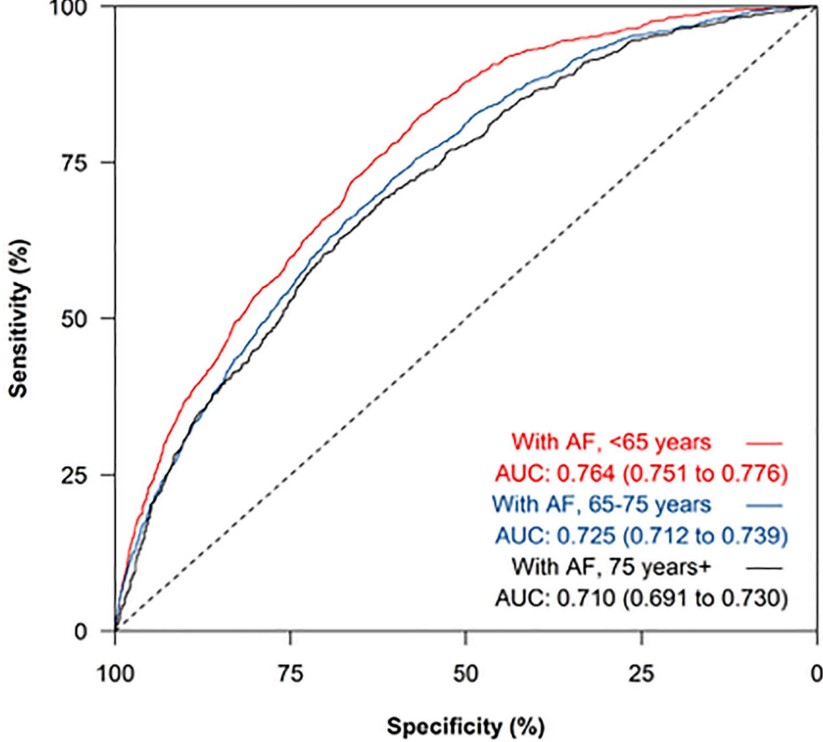

**Fig 3. Receiver operating characteristic curve of NT-proBNP for heart failure diagnosis among people with atrial fibrillation, sub-categorized by age group.** Legend. AUC, area under the curve with 95% confidence interval reported in brackets.

(S2–S7 Tables). This contrasted to people without AF, where the sensitivity and NPV decreased across increasing categories of BMI.

Overall, the diagnostic accuracy of NT-proBNP for HF among people with and without pre-existing AF was similar when comparing by sex (S8 and S9 Tables).

## Potential referral rates between thresholds

Within this cohort, 17,403 people tested had pre-existing AF. An NT-proBNP referral threshold of 125 pg/mL among people with AF would have resulted in 89.7% (n = 15,614) being referred for specialist assessment, with 11,494 false positive results (66.0% of those with AF tested) (Table 2). Setting the threshold at 400 pg/mL in people with AF would result in 18.3% (n = 3,190) fewer referrals, but with a 7.4% (n = 236) increase in the number of people with a missed or delayed diagnosis of HF (S10 Table). Increasing the threshold to 660 pg/mL in people with AF would mean 29.2% (n = 5,073) fewer referrals than the 125 pg/mL threshold and a reduction in false positive to 6,959 (40.0% of those with AF tested) (Table 2) a 26% reduction in people with AF receiving a false positive test result. However, although the NPV remains >90% at the 660 pg/mL threshold among people with AF, using this higher threshold would result in a 10.6% (n = 538) increase in the proportion of those tested having a missed or delayed HF diagnosis.

Among people without AF at a threshold of 125 pg/mL 49.7% (n=68,552) of people who had an NT-proBNP test would have been referred for further assessment of HF and 14.1% (n=9,681) would have had the diagnosis confirmed. Increasing the NT-proBNP threshold to 400 pg/mL among people without AF would result in a 29.9% (n=41,275) reduction in

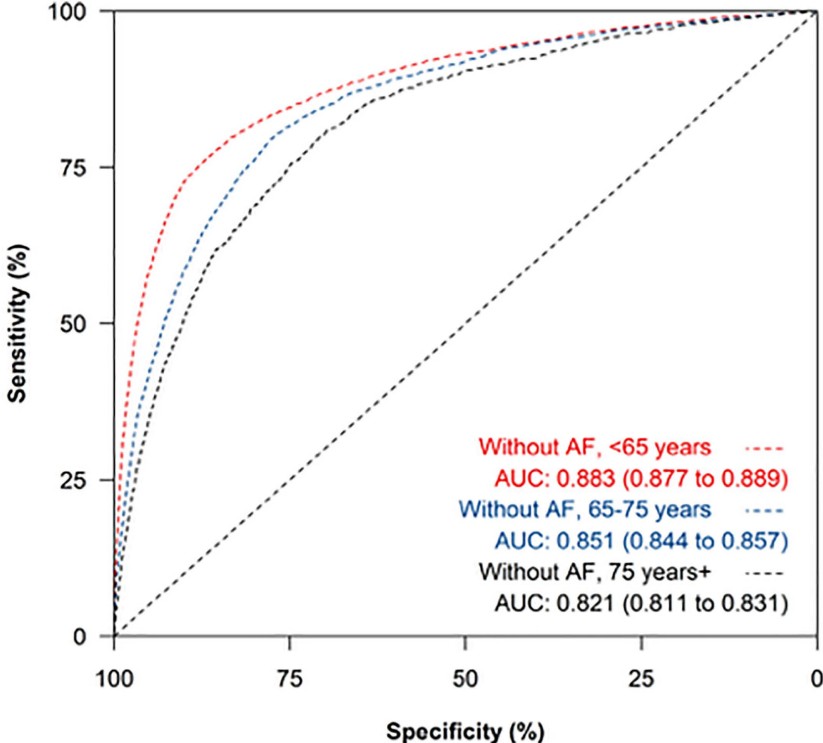

**Fig 4. Receiver operating characteristic curve of NT-proBNP for heart failure diagnosis among people without atrial fibrillation, sub-categorized by age group.** Legend. AUC, area under the curve with 95% confidence interval reported in brackets.

the proportion of people tested being referred for specialist assessment but with a 4% increase ($n$ = 1,652) in missed or delayed HF diagnoses (S10 Table).

## Discussion

In this primary care cohort, the prevalence of HF was more than double among people with pre-existing AF (23.9%) compared to without AF (9.4%). However, median NT-proBNP levels were elevated among people with AF above recommended referral thresholds, irrespective of a subsequent HF diagnosis. The median NT-proBNP was 1,852 pg/mL (IQR [974, 3,459] pg/mL) among people with AF who were subsequently diagnosed with HF but still 724 pg/mL (IQR [245, 1,535] pg/mL) where HF was not diagnosed. Overall, NT-proBNP was better at discriminating for HF among people without AF (AUC = 0.877, 95% CI [0.873, 0.881]) than with AF (AUC = 0.743, 95% CI [0.735, 0.751]). Among people with established AF, the current ESC NT-proBNP referral threshold of 125 pg/mL is an excellent rule-out test for HF with a high NPV (97.3%, 95% CI [96.5, 98.0]) and sensitivity (98.8%, 95% CI [98.5, 99.1]) but most patients (89.7%) will have a raised result and be referred for specialist assessment. Increasing the threshold to 660 pg/mL in people with AF would lead to a 26% reduction in false positive results, whilst retaining the NPV > 90%, albeit with a 10% increase in false negative results.

Diagnostic accuracy of NT-proBNP among people with AF was similar between sexes and by BMI category, but the test's NPV decreased with increasing age. Sensitivity of NT-proBNP was relatively low (70.0%, 95% CI [68.7, 71.4]) among younger people without AF.

A prospective study of 409 patients with AF admitted to the Emergency Department reported that the optimal cut-off for excluding HF among people with AF in the acute setting was 739 ng/L with a NPV of 98% and sensitivity of 99% [17].

PLOS Medicine

**Table 3. Diagnostic test accuracy parameters for the diagnosis of HF using NT-proBNP level by age group at NICE and ESC referral thresholds for those with pre-existing atrial fibrillation.**

| NT-proBNP threshold (pg/mL) | Age <65 years (n=7,499) | | | | Age 65–74 years (n=6,873) | | | | Age 75+ years (n=3,031) | | | |
|---|---|---|---|---|---|---|---|---|---|---|---|---|
| | ≥125 | ≥400 | ≥660 | ≥2,000 | ≥125 | ≥400 | ≥660 | ≥2,000 | ≥125 | ≥400 | ≥660 | ≥2,000 |
| Prevalence % (95% CI) | 19.7 (18.9–20.7) | 19.7 (18.9–20.7) | 19.7 (18.9–20.7) | 19.7 (18.9–20.7) | 26.1 (25.1–27.2) | 26.1 (25.1–27.2) | 26.1 (25.1–27.2) | 26.1 (25.1–27.2) | 29.5 (27.8–31.1) | 29.5 (27.8–31.1) | 29.5 (27.8–31.1) | 29.5 (27.8–31.1) |
| TP, n | 1,459 | 1,361 | 1,222 | 560 | 1,776 | 1,673 | 1,555 | 881 | 885 | 850 | 805 | 503 |
| FN, n | 22 | 120 | 259 | 921 | 18 | 121 | 239 | 913 | 8 | 43 | 88 | 390 |
| FP, n | 4,885 | 3,374 | 2,616 | 646 | 4,585 | 3,521 | 2,935 | 1,056 | 2,024 | 1,645 | 1,408 | 576 |
| TN, n | 1,133 | 2,644 | 3,402 | 5,372 | 494 | 1,558 | 2,144 | 4,023 | 114 | 493 | 730 | 1,562 |
| Sensitivity % (95% CI) | 98.5 (97.8, 99.1) | 91.9 (90.4, 93.2) | 82.5 (80.5, 84.4) | 37.8 (35.3, 40.3) | 99.0 (98.4, 99.4) | 93.3 (92, 94.4) | 86.7 (85 to 88.2) | 49.1 (46.8, 51.4) | 99.1 (98.2, 99.6) | 95.2 (93.6, 96.5) | 90.1 (88.0, 92.0) | 56.3 (53.0, 59.6) |
| Specificity % (95% CI) | 18.8 (17.8, 19.8) | 43.9 (42.7, 45.2) | 56.5 (55.3, 57.8) | 89.3 (88.5, 90.0) | 9.7 (8.9, 10.6) | 30.7 (29.4, 32) | 42.2 (40.8–43.6) | 79.2 (78.1, 80.3) | 5.3 (4.4, 6.4) | 23.1 (21.3, 24.9) | 34.1 (32.1, 36.2) | 73.1 (71.1, 74.9) |
| PPV % (95% CI) | 23.0 (22.0, 24.1) | 28.7 (27.5, 30.1) | 31.8 (30.4, 33.3) | 46.4 (43.6, 49.3) | 27.9 (26.8, 29.0) | 32.2 (30.9, 33.5) | 34.6 (33.2–36) | 45.5 (43.2, 47.7) | 30.4 (28.8, 32.1) | 34.1 (32.2, 36) | 36.4 (34.4, 38.4) | 46.6 (43.6, 49.6) |
| NPV % (95% CI) | 98.1 (97.1, 98.8) | 95.7 (94.8, 96.4) | 92.9 (92.0, 93.7) | 85.4 (84.5, 86.2) | 96.5 (94.5, 97.9) | 92.8 (91.5, 94.0) | 90 (88.7–91.1) | 81.5 (80.4, 82.6) | 93.4 (87.5, 97.1) | 92 (89.3, 94.1) | 89.2 (86.9, 91.3) | 80 (78.2, 81.8) |
| LR+ (95% CI) | 1.21 (1.20, 1.23) | 1.64 (1.60, 1.68) | 1.9 (1.83, 1.97) | 3.52 (3.19, 3.88) | 1.1 (1.09, 1.11) | 1.35 (1.32, 1.38) | 1.5 (1.46–1.55) | 2.36 (2.20, 2.54) | 1.05 (1.03, 1.06) | 1.24 (1.20, 1.27) | 1.37 (1.32, 1.42) | 2.09 (1.91, 2.29) |
| LR− (95% CI) | 0.08 (0.05, 0.12) | 0.18 (0.15, 0.22) | 0.31 (0.28, 0.35) | 0.7 (0.67, 0.73) | 0.1 (0.06, 0.16) | 0.22 (0.18, 0.26) | 0.32 (0.28–0.36) | 0.64 (0.61, 0.67) | 0.17 (0.08, 0.34) | 0.21 (0.15, 0.28) | 0.29 (0.23, 0.35) | 0.60 (0.55, 0.65) |
| DOR (95% CI) | 15.27 (10.23, 24.11) | 8.88 (7.35, 10.83) | 6.13 (5.32, 7.09) | 5.05 (4.43, 5.78) | 10.54 (6.77, 17.57) | 6.11 (5.05, 7.45) | 4.75 (4.11–5.51) | 3.68 (3.28, 4.12) | 6.11 (3.17, 13.76) | 5.9 (4.33, 8.26) | 4.73 (3.75, 6.04) | 3.50 (2.97, 4.12) |

**Abbreviations:** DOR, diagnostic odds ratio, FN, false negatives, FP, false positives, LR, likelihood ratio, N, number, NPV, negative predictive value, PPV, positive predictive value, TN, true negatives, TP, true positives.

The standard ESC recommended NT-proBNP threshold in the acute setting is 300 ng/L [1]. However, these studies did not report on the diagnostic accuracy of NT-proBNP for HF among people with AF among a primary care cohort to inform whether similar adjustments should be made in the community setting.

The ESC HF Association consensus statement highlights how using a single NT-proBNP threshold may result in unnecessary referrals and investigation among patients groups where levels are known to be elevated [4]. Age-specific cut-offs are therefore recommended with further adjustment to account for BMI, kidney failure and AF. However, it is acknowledged that these suggestions are 'based more on expert opinion rather than on strong evidence and should be refined as more information becomes available' [4]. Our results demonstrate that the presence of AF impacts upon the diagnostic accuracy of NT-proBNP to a greater extent than age or BMI so these recommendations may need to reviewed.

Most prior diagnostic accuracy studies of NT-proBNP have included relatively small numbers of patients with AF, e.g., <1,000 [18–20]. Larger studies have typically reported on median NT-proBNP ranges among the general population [6,21]. To our knowledge, this is the largest diagnostic accuracy study of NT-proBNP for HF among people with AF. Our

study includes everybody who had an NT-proBNP test ordered within a large primary care cohort over a 14 year period, meaning the results are directly applicable to general practice. NT-proBNP tests tend to be ordered by clinicians in primary care for patients with suspected heart failure, but there may be some selection bias in terms of who clinicians selected for testing based on presenting symptoms. Nonetheless, our results inform the interpretation of NT-proBNP among those undergoing testing.

The size of the cohort is important because it allows us to provide an accurate estimate of the prevalence of HF among people with and without AF, which are essential to report the NPV and PPV. Laboratory test codes are accurately recorded in electronic health records meaning the cohort could be reliably identified [22].

One limitation of the analysis is the reliance on observational data. Patients with an NT-proBNP result <400 pg/mL would not have met the NICE referral threshold for further evaluation of HF. This means the total number of patients with an NT-proBNP<400 pg/mL who had a false negative result is likely to be under-estimated and the sensitivity results over-estimated. However, we did observe people with a NT-proBNP result <400 pg/mL subsequently diagnosed with HF, suggesting that clinicians use a variety of approaches to determine which patients need further investigation and do not rely on NT-proBNP alone. Some patients may not have been initially referred but were subsequently further investigated after a follow-up appointment or following an acute admission. Furthermore, clinicians making the diagnosis of HF are likely to have had access to the NT-proBNP result (index test), which may have influenced the diagnostic process. Some patients may have been referred for suspected heart failure but still not assessed within the six month time window we used leading to erroneous false negative NT-proBNP results. However, given UK guidelines recommend people with a raised NT-proBNP are assessed within six weeks of referral we think this would be unlikely to apply to so many patients it would alter the summary findings.

Coding of HF based on the LVEF is limited within the dataset, precluding an analysis of the relative diagnostic accuracy of NT-proBNP for HF with reduced ejection fraction (HFrEF) or HFpEF. Previous studies have reported that NT-proBNP may be better at detecting patients with HFrEF than HFpEF [23], and that up to one third of patients with HFpEF may have NT-proBNP levels <100 pg/mL [24]. However, clinicians referring patients with suspected HF will not know whether a patient has HFrEF or HFpEF at the time of diagnosis, meaning that diagnostic thresholds in the community need to be kept at a uniform level across categories of HF, reflected in our approach to the current analysis.

We did not extract data for cardiovascular medication, heart rate or kidney function as these parameters were not considered critical to the central study design. Furthermore it is not possible to tell whether patients who had been previously diagnosed with AF were actually in AF at the time of their NT-proBNP test. However, each of these factors could be important when interpreting NT-proBNP levels in practice and in determining the optimal threshold to use for suspected heart failure. Further research could involve a dedicated analysis considering these variables.

Diagnosing HF in patients with AF can be challenging given the overlap of symptoms and presence of co-morbidities that can lead to diagnostic overshadowing. Interpreting NT-proBNP results in people with AF is difficult because one must consider both the fact HF is more prevalent among people with AF but also AF will increase NT-proBNP levels. As a result, NT-proBNP provides a less informative result for suspected HF among people with AF, compared to those without. Future research could explore to what extent NT-proBNP thresholds can simultaneously account for multiple other factors such as age, BMI, kidney function and current treatment to improve the predictive performance for suspected heart failure.

A NT-proBNP threshold of 125 pg/mL among people with AF is a reliable 'rule out' test in primary care. Echocardiogram offers a risk-free and non-invasive next step in the diagnostic pathway meaning the potential harms to an individual patient of a positive NT-proBNP are minimal. Early identification of HF may be of particular importance among people with AF, in contrast to age, because it directly impacts upon treatment decisions. An early rhythm-based approach to treatment among people with AF and HFrEF using catheter ablation can improve outcomes, such as reducing hospitalizations for HF and improving left ventricular systolic function, quality of life and survival [25–28].

However, most patients with AF will have an NT-proBNP greater than 125 pg/mL meaning this threshold does not discriminate accurately for HF and at that threshold most patients will be referred for further assessment, with significant resource implications for health services. We suggest a higher threshold of 660 pg/mL is considered to account for the fact that AF is associated with raised NT-proBNP levels to help target echocardiogram resources at those most likely to have HF. A NT-proBNP threshold of 400 pg/mL in people with AF would maintain a test sensitivity (93.2%, 95% CI [92.4, 93.9]) and NPV (94.3%, 95% CI [93.6, 94.9]) above 90% but reduce referrals by 18.3% with a 7.4% increase in missed or delayed diagnoses compared to the 125 pg/mL threshold. Given elevated NT-proBNP levels have prognostic significance, this approach would also be likely to prioritize the highest risk patients for earlier diagnostic assessment.

Half of people without AF would have a negative NT-proBNP at the 125 pg/mL threshold with a high NPV (98.9%, 95% CI [98.9, 99.0]) and sensitivity (92.9%, 95% CI [92.4, 93.4]) for excluding HF. The ESC threshold of 125 pg/mL may therefore be most appropriate for people without AF.

Ultimately the threshold may be determined by healthcare system factors, such as access to echocardiography, where higher NP levels could support a fast-track referral. Implementation studies could help determine the clinical and cost impact of changing the NT-proBNP threshold. Clinical judgement will remain important when interpreting NT-proBNP results for individual patients with AF to take account of other factors, such as the patient's age, any concurrent illness, current treatment and heart rate as well as the index of suspicion as to a suspected heart failure diagnosis.

People with AF are known to be at high-risk of developing HF but NT-proBNP discriminates less accurately for HF among people with AF compared to those without AF. A low NT-proBNP referral threshold among people with AF would minimize missed or delayed diagnosis but would also have significant resource implications. Conversely, raising the referral threshold for a positive test could help target investigations at those people with AF most likely to have HF. Implementation studies could further be done to evaluate the comparative cost and clinical-effectiveness of different NT-proBNP thresholds.

## Supporting information

**S1 Table. Diagnostic test accuracy parameters for the diagnosis of HF using NT-proBNP level by age group at NICE and ESC referral thresholds for those without pre-existing atrial fibrillation.**
(PDF)

**S2 Table. Diagnostic test accuracy parameters for the diagnosis of HF using NT-proBNP level among people with underweight (BMI < 20 kg/m$^2$) at NICE and ESC referral thresholds based on presence of pre-existing atrial fibrillation.**
(PDF)

**S3 Table. Diagnostic test accuracy parameters for the diagnosis of HF using NT-proBNP level among people with healthy weight (BMI 20–25 kg/m$^2$) at NICE and ESC referral thresholds based on presence of pre-existing atrial fibrillation.**
(PDF)

**S4 Table. Diagnostic test accuracy parameters for the diagnosis of HF using NT-proBNP level among people with overweight (BMI 25–30 kg/m$^2$) at NICE and ESC referral thresholds based on presence of pre-existing atrial fibrillation.**
(PDF)

**S5 Table. Diagnostic test accuracy parameters for the diagnosis of HF using NT-proBNP level among people with obesity stage 1 (BMI 30–35 kg/m$^2$) at NICE and ESC referral thresholds based on presence of pre-existing atrial fibrillation.**
(PDF)

**S6 Table. Diagnostic test accuracy parameters for the diagnosis of HF using NT-proBNP level among people with obesity stage 2 (BMI 35–40 kg/m$^2$) at NICE and ESC referral thresholds based on presence of pre-existing atrial fibrillation.**
(PDF)

**S7 Table. Diagnostic test accuracy parameters for the diagnosis of HF using NT-proBNP level among people with obesity stage 3 (BMI ≥ 40 kg/m2) at NICE and ESC referral thresholds based on presence of pre-existing atrial fibrillation.**
(PDF)

**S8 Table. Diagnostic test accuracy parameters for the diagnosis of HF using NT-proBNP level by sex at NICE and ESC referral thresholds for those without pre-existing atrial fibrillation.**
(PDF)

**S9 Table. Diagnostic test accuracy parameters for the diagnosis of HF using NT-proBNP level by sex at NICE and ESC referral thresholds for those with pre-existing atrial fibrillation.**
(PDF)

**S10 Table. Proportion of patients undergoing NT-proBNP testing who would be referred for further assessment and have a diagnosis of heart failure confirmed, comparing between different NT-proBNP thresholds and based on presence of pre-existing atrial fibrillation.**
(PDF)

**S1 Appendix. Codes used to identify people undergoing NT-proBNP testing and diagnosed with heart failure or atrial fibrillation within the study.**
(PDF)

**S1 Fig. ROC curve of NT-proBNP among people with pre-existing atrial fibrillation, comparing between categories of body mass index.**
(PDF)

**S2 Fig. ROC curve of NT-proBNP among people with pre-existing atrial fibrillation, comparing between categories of body mass index.**
(PDF)

**S1 Checklist. Completed Standards for Reporting of Diagnostic Accuracy (STARD) guideline.**
(PDF)

## Acknowledgments

This work uses data provided by patients and collected by the NHS as part of their care and support and would not have been possible without access to this data. The NIHR recognizes and values the role of patient data, securely accessed and stored, both in underpinning and leading to improvements in research and care. The views expressed are those of the authors and not necessarily those of the NHS, the NIHR, or the Department of Health and Social Care.

## Author contributions

**Conceptualization:** Nicholas R. Jones, Clare J. Taylor.

**Formal analysis:** Nicholas R. Jones, Kathryn S. Taylor, José M. Ordóñez-Mena, Clare J. Taylor.

**Funding acquisition:** F. D. Richard Hobbs, Clare J. Taylor.

**Investigation:** Nicholas R. Jones, Kathryn S. Taylor, José M. Ordóñez-Mena.

**Methodology:** Nicholas R. Jones, Kathryn S. Taylor, José M. Ordóñez-Mena, F. D. Richard Hobbs, Clare J. Taylor.

**Project administration:** F. D. Richard Hobbs, Clare J. Taylor.

**Software:** Nicholas R. Jones, José M. Ordóñez-Mena.

**Supervision:** F. D. Richard Hobbs, Clare J. Taylor.

**Validation:** Kathryn S. Taylor, Clare R. Goyder, Clare J. Taylor.

**Visualization:** Nicholas R. Jones, José M. Ordóñez-Mena, F. D. Richard Hobbs.

**Writing – original draft:** Nicholas R. Jones.

**Writing – review & editing:** Kathryn S. Taylor, José M. Ordóñez-Mena, Clare R. Goyder, F. D. Richard Hobbs, Clare J. Taylor.

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
