## [Editor Report · Decision Letter 0]

30 Jan 2025

Dear Dr Jones,

Thank you for submitting your manuscript entitled "NT-proBNP testing for heart failure diagnosis in people with atrial fibrillation: diagnostic accuracy study" for consideration by PLOS Medicine.

Your manuscript has now been evaluated by the PLOS Medicine editorial staff and I am writing to let you know that we would like to send your submission out for external peer review.

Please re-submit your manuscript within two working days, i.e. by Feb 03 2025.

Feel free to email me at atosun@plos.org or us at plosmedicine@plos.org if you have any queries relating to your submission.

Kind regards,

Alexandra Tosun, PhD

Associate Editor

PLOS Medicine

---

## [Decision Letter · Decision Letter 1]

18 Jun 2025

Dear Dr Jones,

Many thanks for submitting your manuscript "NT-proBNP testing for heart failure diagnosis in people with atrial fibrillation: diagnostic accuracy study" (PMEDICINE-D-25-00239R1) to PLOS Medicine. The paper has been reviewed by subject experts and a statistician; their comments are included below and can also be accessed here: [LINK]

As you’ll see, the reviewers provided valuable feedback and raised several concerns about the study. After discussing the paper with the editorial team and an guest academic editor with relevant expertise, I'm pleased to invite you to revise the paper in response to the reviewers' comments. We plan to send the revised paper to some or all of the original reviewers, and we cannot provide any guarantees at this stage regarding publication.

We ask that you submit your revision by Jul 09 2025. However, if this deadline is not feasible, please contact me by email, and we can discuss a suitable alternative.

Don't hesitate to contact me directly with any questions (atosun@plos.org).

Best regards,

Alexandra

Alexandra Tosun, PhD

Senior Editor

PLOS Medicine

atosun@plos.org

Comments from the academic editor:

The topic is interesting and providing more precise cutoffs for the diagnosis of HF is valuable, considering population heterogeneity where factors such as AF and age play significant roles. The use of a large-scale, real-world UK database further strengthens the reliability of the data.

However, the key issue is that BNP/NT-proBNP is primarily used as a rule-out biomarker for HF diagnosis. The current manuscript does not adequately address this perspective, focusing instead on general predictive performance as measured by AUC. Additionally, there are important questions raised by all other reviewers regarding the criteria used for population selection, particularly in how AF and HF are defined within the database.

Comments from the reviewers:

Reviewer #1: This article is on the effectiveness of NtproBNP as a diagnostic marker of heart failure in those with and without heart failure. Other reviewers will be better placed to comment on the clinical aspects of the study and its potential impact. However, in my opinion the statistical methods are appropriate given the available data, and the conclusions taken from the analysis are reasonable. As a result I only have minor comments on the article:

- In the tables it is being shown that there are thousands of individuals (2588 with AF and 9782 without) with HF diagnosed 6 or more weeks after the NtproBNP test that are nonetheless being considered as 'without heart failure'. As far as I can tell the rationale for this time cut off is not discussed within the report and also the implications that a reasonably high proportion (~5-10%) of this population do go on to develop heart failure. If known, it potentially also would be useful to report on more exact measures of time (months, years) the status is confirmed as opposed to just reporting it more than six weeks.

- Although the sensitivity, specificity and AUC using previously reported thresholds are reported, I don't think there is any proposal as to what in theory the best thresholds could be for this AF population using the available data under different sensitivity/specificity considerations. To some extent this can be read from figure 1, but could be useful to have exact values reported for different scenarios, eg if there is a need for >99% sensitivity for a screening tool what threshold and specificity would result.

- Other than in the conclusions, there is no description as to why the high sensitivity/low specificity is appropriate for this context in using NtProBNP as an initial screening tool.

- Figure 2 is quite cluttered and could potentially be split into two separate plots or otherwise laid out clearer.

- Although often standard text for statistical methods sections, it isn't clear within the context of the paper what p-values are actually being assessed with the "5% threshold to define significance" statement being made. No p-values are reported throughout, presumably as comparing to zero would be largely irrelevant, and the more meaningful 95% confidence intervals are already described elsewhere in methods.

Reviewer #2: NT-proBNP is often used to assess suspected heart failure (HF), but levels are elevated in atrial fibrillation (AF), making diagnosis difficult. Atrial fibrillation patients are at higher risk of HF, but an elevated NT-proBNP level in this group does not always indicate HF, leading to diagnostic uncertainty. NT-proBNP has better diagnostic accuracy for HF in patients without atrial fibrillation (AUC = 0.877) than in those with atrial fibrillation (AUC = 0.743). Large cohort (n=155,347), use of UK primary care data linked to secondary care. Real-world data from a representative population over 14 years (2004–2018).

Should NT-proBNP thresholds be routinely adjusted for patients with AF in primary care, or should clinicians rely on their clinical judgment?

How might the inclusion of other biomarkers or clinical parameters improve diagnostic accuracy in patients with atrial fibrillation?

How does the lack of left ventricular ejection fraction (LVEF) data limit the interpretation of NT-proBNP performance in different HF subtypes?

Since NT-proBNP values are also influenced by renal function and medications, how could the inclusion of these factors have altered the results?

Reviewer #3: The aim of this study was to assess the diagnostic accuracy of NT-proBNP for HF in a primary care cohort (>155k) of people with AF, overall and by age, sex and BMI.

They conclude that NT-proBNP discriminates more accurately for HF among people without AF than with AF. No surprises here.

Higher natiuretic peptides in AF are well reported, also affected by age, renal function, underlying renal dysfunction, infection etc - also some racial differences too.

A higher referral threshold could be considered in AF to account for higher median NT-proBNP levels. Agreed. This has been advocated in prior papers.

NT-proBNP (and many biomarkers) may be better as a rule out, rather than rule in. The nonspecificity and dynamic changes in levels over time (not just a single 'one off' value) are other limitations of these biomarkers which may simply reflect a sick individual. The present analysis also does not properly/definitively distinguish HFrEF, HFpEF, HRmrEF etc as LVEF only recorded in 836

The analysis also depends on CPRD, which means that the data are based on subjects who had a blood measurement 'because of something' eg comorbidity or GP suspicion of HF (well, breathlessness) - rather than an unselected population based screening

'Absence of a HF code was assumed to mean absence of a diagnosis.' How accurate is this approach? Did you test the internal and external validity here?

Reviewer #4: The manuscript 'NT-proBNP testing for heart failure diagnosis in people with atrial fibrillation: diagnostic accuracy study' addresses an important issue with regards to how to use/interpret NT-proBNP testing in patients with suspicion of HF in the setting of AF. The paper is well written and addresses an difficult aspect of a widely use test in health care overall and primary care in particular. The obvious strength of the current manuscript is the big sample size with more than 150 000 patients included in the analysis. Even though it remains a challenge to decide what the appropriate cut-off for NT-proBNP is in the population in general and AF patients in particular, this study could contribute to a further understanding of the challenges associated with the use of natriuretic peptides in the setting of HF and AF. However, there are some questions remaining after having gone through the manuscript.

1. On important issue is the diagnosis of AF. As I understand it, the patients with AF was categorized to this entity based on the diagnosis of AF in EHR? I suppose that this means that it is sufficient with a diagnosis during the course up to study start, meaning that we have no idea whether the patients were in AF at the time of NT-proBNP testing? If so, I think this is an important limitation when the results are interpreted, where a patients with ongoing fast AF is quite different from a patient with rare episodes af AF and that are in SR during testing. What are the authors thoughts about this? This is an important issue when it comes to implementing the results in clinical practice.

2. I think the paper could benefit from a figure showing the flow chart of patient inclusion.

3. The authors state that kidney disease is a factor that could complicate interpretation of NT-proBNP levels, and in the discussion section list this at one factor to take into account when interpretating the results. Even so, it is stated in the discussion that "....kidney function were not considered critical to the central study design"? Is data regarding kidney function available and if so, have they been tested in the statistical modelling?

4. In the methods section, are the codes for the different conditions specified anywhere?

5. In the results section, it is stated that there were only 836 patients with LVEF recorded, what is the reason for this?

6. Furthermore, in the same area, it is stated that the median LVEF was 48% among people with AF diagnosed with HF compared to 40 % in those not (!) diagnosed with HF? This seem strange.

Reviewer #5: This reviewer can see two versions of the manuscript. However, there is no communication between the authors and reviewers. Thus, this reviewer can only provide comments as for an original submission.

The authors utilized real world data from primay and secondary care to look at the dignostic value of NT-proBNP for heart failure in the presence and absence of atrial fibrillation. The primary finding was that NT-pro-BNP behaved poorly in the diagnosis of heart failure in patients with atrial fibrillation. The analysis was well done. The results are well presented. This reviewer only has several minor comments.

1. In the presence of atrial fibrillation, NT-proBNP behaved poor in general, and even poorer with age advancing, not typically in those over 75 years. The authors might make the latter point instead of reporting the poorer results in those over 75 years of age. In fact, the AUC was very similar in those of 65-74 years old.

2. The legend inside to Figure 2 is difficult to read. The authors may consider to make some space between those with atrial fibrillation and those without atrial fibrillation similarly as they did for Figure 1.

3. The tables are generally big and busy. If possible, the authors may consider to move some of them, for instance, the one between men and women, to supplemental materials, or simplify the contents, for instance the two "total" columns in table 1.

Reviewer #6: This study evaluated the diagnostic accuracy of NT-proBNP for heart failure (HF) in patients with and without atrial fibrillation (AF) using the large-scale CPRD GOLD and Aurum databases in UK with 155,347 patients. The authors proposed considering higher AF-specific NT-proBNP thresholds for HF due to elevated baseline levels. The authors addressed a clinically relevant question, i.e. NT-proBNP's utility in HF diagnosis for patients with AF. The large sample size and real-world data linkage are notable strengths. However, several methodological and interpretive issues are listed below which warrant discussion:

1. Study population selection: this study included patients tested for NT-proBNP from the database. This should be clarified whether these corresponded to the suspected HF cases per guidelines, e.g., NICE 2003 (considering that their study started from 2004) recommends natriuretic peptide testing for clinical suspicion using BNP or NT-proBNP where available. Could echocardiograms or other confirmatory tests alone be performed for HF diagnosis? This impacts the interpretation of "diagnostic accuracy," as NT-proBNP is rarely used in isolation.

2. Statistics for rule-out utility: the emphasis on AUC may be misleading for evaluating a "rule-out biomarker". For exclusion, I believe that NPV and sensitivity are more important. As in Table 2 shows, comparable NPVs for AF vs non-AF patients at 125 pg/mL (97.3% vs. ~98.9%) and 400 pg/mL (94.3% vs. 97.8%). Sensitivity remains high in AF patients (98.8% and 93.2%) at 125 and 400 pg/mL respectively. In my opinion, retaining thresholds for AF patients (e.g., 125 pg/mL) would minimally increase false positives while safeguarding against missed diagnoses, a trade-off favoring safety in real-world practice given AF's around 10% prevalence in the cohort.

3. Outcome definition: the 6-month HF diagnosis endpoint diverges from typical acute diagnostic workflows (where NT-proBNP guides the immediate decisions for further evaluations). Could elevated NT-proBNP in AF have led to earlier HF detection during follow-up, inflating accuracy?

---

* Please upload any figures associated with your paper as individual TIF or EPS files with 300dpi resolution at resubmission; please read our figure guidelines for more information on our requirements: http://journals.plos.org/plosmedicine/s/figures. While revising your submission, please upload your figure files to the PACE digital diagnostic tool, https://pacev2.apexcovantage.com/. PACE helps ensure that figures meet PLOS requirements. To use PACE, you must first register as a user. Then, login and navigate to the UPLOAD tab, where you will find detailed instructions on how to use the tool. If you encounter any issues or have any questions when using PACE, please email us at PLOSMedicine@plos.org.

* The funding statement should include: specific grant numbers, initials of authors who received each award, URLs to sponsors’ websites. Also, please state whether any sponsors or funders (other than the named authors) played any role in study design, data collection and analysis, the decision to publish, or preparation of the manuscript. If they had no role in the research, include this sentence: “The funders had no role in study design, data collection and analysis, decision to publish, or preparation of the manuscript.”

* Data availability: If the data are not freely available, please describe briefly the ethical, legal, or contractual restriction that prevents you from sharing it. Please also include an appropriate contact (web or email address) for inquiries. Please note that this cannot be a study author.

* Ethics: Please provide details on consent and ensure to provide the ISAC approval number if available.

FIGURES AND TABLES

SUPPLEMENTARY MATERIAL

REFERENCES

STUDY TYPE-SPECIFIC REQUESTS

* Abstract: Please include the study design, population and setting, number of participants, years during which the study took place (enrollment and follow up), length of follow up, and main outcome measures.

* For all observational studies, in the manuscript text, please indicate: (1) the specific hypotheses you intended to test, (2) the analytical methods by which you planned to test them, (3) the analyses you actually performed, and (4) when reported analyses differ from those that were planned, transparent explanations for differences that affect the reliability of the study's results. If a reported analysis was performed based on an interesting but unanticipated pattern in the data, please be clear that the analysis was data driven.

* Please state in the Methods section whether the study had a prospective protocol or analysis plan. If a prospective analysis plan (from your funding proposal, IRB or other ethics committee submission, study protocol, or other planning document written before analyzing the data) was used in designing the study, please include the relevant document(s) with your revised manuscript as a Supporting Information file to be published alongside your study and cite it in the Methods section. A legend for this file should be included at the end of your manuscript. If no such document exists, please make sure that the Methods section transparently describes when analyses were planned, and when/why any data-driven changes to analyses took place. Changes in the analysis, including those made in response to peer review comments, should be identified as such in the Methods section of the paper, with rationale.

* Please ensure that the study is reported according to the STARD guideline (https://www.equator-network.org/reporting-guidelines/stard/) and include the completed STARD checklist as Supporting Information. Please add the following statement, or similar, to the Methods: "This study is reported as per the Standards for Reporting of Diagnostic Accuracy (STARD) guideline (S1 Checklist)." When completing the checklist, please use section and paragraph numbers, rather than page numbers.

* Please structure your Abstract according to STARD for Abstracts (https://www.equator-network.org/reporting-guidelines/stard-abstracts/).

* Please structure the Methods section using the following sub-headings: Study design, Participants, Test methods, Analysis.

* Please include a diagram to describe the flow of participants through the study (typically figure 1).

---

## [Decision Letter · Decision Letter 2]

1 Aug 2025

Dear Dr. Jones,

Thank you very much for re-submitting your manuscript "NT-proBNP testing for heart failure diagnosis in people with atrial fibrillation: diagnostic accuracy study" (PMEDICINE-D-25-00239R2) for review by PLOS Medicine.

I have discussed the paper with my colleagues and the academic editor and it was also seen again by 4 reviewers. I am pleased to say that provided the remaining editorial and production issues are dealt with we are planning to accept the paper for publication in the journal.

We like you to address the remaining comments from reviewer #1. Furthermore, we would also like you to address the remaining concerns from the Academic Editor. Furthermore, there are a number of editorial issues that need to be addressed, which are listed at the end of this email. Any accompanying reviewer attachments can be seen via the link below. Please take these into account before resubmitting your manuscript:

We look forward to receiving the revised manuscript by Aug 08 2025 11:59PM.   

Sincerely,

Suzanne de Bruijn

On behalf of

Alexandra Tosun, PhD

Senior Editor 

PLOS Medicine

plosmedicine.org

Requests from Editors:

-The suggestion that 26% of people with AF that are currently classified as false positives could be avoided while maintaining the acceptable NPV, merits greater visibility and should be highlighted further in the MS, as well as the abstract and author summary, to emphasize the clinical implications of this finding.

-Please provide more detail and discussion on potential selection bias. It would be helpful to clarify how this population compares to the real-world population, and a brief discussion of this topic would help the generalizability of the findings.

-Academic Editor: I would suggest modifying the prevalence in Table 2 to be more specific. it would be useful to include the prevalence of HF according to different cutoffs. If it’s the overall prevalence of HF, it could be listed along with the group title.

GENERAL EDITORIAL REQUESTS

* Please confirm that your title complies with to PLOS Medicine's style. Your title must be nondeclarative and not a question. It should begin with main concept if possible. "Effect of" should be used only if causality can be inferred, i.e., for an RCT. Please place the study design ("A randomized controlled trial," "A retrospective study," "A modelling study," etc.) in the subtitle (ie, after a colon).

* Please confirm that your abstract complies with our requirements, including format (three sections: Background, Methods and Findings, and Conclusions) and providing all the information relevant to this study type https://journals.plos.org/plosmedicine/s/submission-guidelines#loc-abstract

* Please ensure that the Introduction ends with a clear description of the study question or hypothesis.

* Please ensure that all abbreviations are defined at first use throughout the text.

* Please confirm that all numbers presented in the abstract are present and identical to numbers presented in the main manuscript text.

GENERAL

* Please remove the 'conclusions' subheading from the discussion. Please also remove any other subheadings from the discussion.

* Statistical reporting: Please revise throughout the manuscript, including tables and figures.

- Please report statistical information as follows to improve clarity for the reader ""22% (95% CI [13,28]; p</=)"".

- Please separate upper and lower bounds with commas instead of hyphens as the latter can be confused with reporting of negative values.

- Please repeat statistical definitions (HR, CI etc.) for each set of parentheses.

FUNDING STATEMENT

* The funding statement should include: specific grant numbers, initials of authors who received each award, URLs to sponsors’ websites. Also, please state whether any sponsors or funders (other than the named authors) played any role in study design, data collection and analysis, the decision to publish, or preparation of the manuscript. If they had no role in the research, include this sentence: “The funders had no role in study design, data collection and analysis, decision to publish, or preparation of the manuscript.”

* It appears that one or more study authors is affiliated with one or more of the agencies that funded the study. Thus, the statement “The funders had no role in study design, data collection and analysis, decision to publish, or preparation of the manuscript” does not apply. Please revise the Financial Disclosure accordingly, as in "[Author name] is [author's role] at [funding agency]. The funders had no other role in study design…..” Specifically, this applies to NRJ.

CODE

* Please state in your data availability statement where the code can be found, with a URL to the GITHUB entry.

COMPETING INTERESTS STATEMENT

* All authors must declare their relevant competing interests per the PLOS policy, which can be seen here: https://journals.plos.org/plosmedicine/s/competing-interests For authors with ties to industry, please indicate whether any of the interests has a financial stake in the results of the current study.

DATA AVAILABILITY

* PLOS Medicine requires that the de-identified data underlying the specific results in a published article be made available, without restrictions on access, in a public repository or as Supporting Information at the time of article publication, provided it is legal and ethical to do so. Please see the policy at

http://journals.plos.org/plosmedicine/s/data-availability and FAQs at http://journals.plos.org/plosmedicine/s/data-availability#loc-faqs-for-data-policy "

"* The Data Availability Statement (DAS) requires revision. For each data source used in your study:

a) If the data are owned by a third party but freely available upon request, please note this and state the owner of the data set and contact information for data requests (web or email address). Note that a study author cannot be the contact person for the data.

b) If the data are not freely available, please describe briefly the ethical, legal, or contractual restriction that prevents you from sharing it. Please also include an appropriate contact (web or email address) for inquiries (again, this cannot be a study author).

ETHICS AND CONSENT

*Thank you for including an ethics statement. However this is not, as mentioned in the metadata, present in the Methods section’. Please ensure the metadata correctly states where the ethics statement can be found.

* Please specify whether informed consent was written or oral. Please ensure that the research complies with the PLOS policy in full: https://journals.plos.org/plosmedicine/s/human-subjects-research#loc-patient-privacy-and-informed-consent-for-publicationNT

FIGURES

* Please number the ‘central illustration’ so it is clear where in the manuscript this fits.

DIAGNOSTIC TESTS

*Did your study have a prospective protocol or analysis plan? Please state this (either way) early in the Methods section.

* Address the extent to which the study population is representative of the population of interest.

Comments from Reviewers:

Reviewer #1: Reviewer 1:

In my previous review of the article i was of the opinion that it was well written with appropriate methods used, and that remains the case for the re-submission. I believe that the authors have made all reasonable attempts to address my own and other reviewer comments, and that the changes have further improved the article including addressing where there had apparently been some uncertainty in the clinical interpretation.

The only minor comment to address from the responses is for my item #5, where I wasn't suggesting that p-values would need to be added as I agree the confidence intervals are more informative, just that it is odd to state the threshold of significance used for p-values when none are being reported in the paper.

Reviewer #3: No additional comments

Reviewer #5: No further comments.

---

## [Editor Report · Decision Letter 3]

26 Aug 2025

Dear Dr Jones, 

On behalf of my colleagues and the Guest Academic Editor, Pei Gao, I am pleased to inform you that we have agreed to publish your manuscript "NT-proBNP testing for heart failure diagnosis in people with atrial fibrillation: diagnostic accuracy study" (PMEDICINE-D-25-00239R3) in PLOS Medicine.

I appreciate your thorough responses to the reviewers' and editors' comments throughout the editorial process. We look forward to publishing your manuscript. Editorially, there are a few remaining points that should be addressed prior to publication. We will carefully check whether the changes have been made. If you have any questions or concerns regarding these final requests, please feel free to contact me at atosun@plos.org.

1) Title: We suggest changing it to: “NT-proBNP testing for heart failure diagnosis in people with atrial fibrillation: A diagnostic accuracy study”

2) Abstract: Please include the study setting.

3) Abstract: ”Among 155,347 people tested (median age 61 years), 17,403 (11.2%) had pre-existing AF. 14,585 (9.4%) people were diagnosed with HF, including 4,168 (23.9%) with AF (median NT-proBNP =1,852pg/mL, IQR [974, 3,459]pg/mL) and 10,417 (7.6%) without AF (1,110pg/mL, IQR [434, 3,108]pg/mL).” – When reporting percentages, please report the numerator and denominator, particularly because the denominators seem to vary for the numbers reported here. Also, note that 10,417/155,347 equals 6.7%, not 7.6% (if denominator correct).

4) Abstract: Please define IQR and CI at first use.

5) Figures: Please add the statistical definition of the numbers in brackets, e.g. 95%CI values.

6) Please include the code availability statement in your data availability statement in the online submission form.

7) STARD checklist: Please replace the page numbers with paragraph numbers per section (e.g. "Methods, paragraph 1"), since the page numbers of the final published paper may be different from the page numbers in the current manuscript.

Before your manuscript can be formally accepted you will need to complete some formatting changes, which you will receive in a follow up email (including the editorial requests above). Please be aware that it may take several days for you to receive this email; during this time no action is required by you. Once you have received these formatting requests, please note that your manuscript will not be scheduled for publication until you have made the required changes.

PRESS

Sincerely, 

Alexandra Tosun, PhD 

Senior Editor 

PLOS Medicine